# A General Scheme of a Branch-and-Bound Approach for the Sensor Selection Problem in Near-Field Broadband Beamforming

**DOI:** 10.3390/s24020470

**Published:** 2024-01-12

**Authors:** Agnieszka Wielgus, Bogusław Szlachetko

**Affiliations:** Department of Acoustics, Multimedia and Signal Processing, Wroclaw University of Science and Technology, Wyb. Wyspianskiego 27, 50-370 Wrocław, Poland; boguslaw.szlachetko@pwr.edu.pl

**Keywords:** sensor selection problem, beamforming, branch and bound

## Abstract

This paper is devoted to the sensor selection problem. A broadband receiver beamforming working in a near-field is considered. The system response should be as close as possible to the desired one, which is optimized in the sense of $L_{2}$ norm. The problem considered is at least NP-hard. Therefore, the branch-and-bound algorithm is developed to solve the problem. The proposed approach is universal and can be applied not only to microphone arrays but also to antenna arrays; that is, the methodology for the generation of consecutive solutions can be applied to different types of sensor selection problems. Next, for a larger microphone array, an efficient metaheuristic algorithm is constructed. The algorithm implemented is a hybrid genetic algorithm based on the ITÖ process. Numerical experiments show that the proposed approach can be successfully applied to the sensor selection problem.

## 1. Introduction

Beamforming is a technique widely used in acoustic signal processing to improve microphone directionality. This technique involves combining many microphones into an array system distributed in space according to specific patterns [1]. Such arrays, called microphone arrays, are particularly effective when there are multiple sound sources and the desired signal must be separated from unwanted noise. Additionally, the microphone array can be used to estimate the direction of arrival of the wave (DOA). An important aspect is the fact that signals can be separated even when they occupy the same frequency band but arrive at the array from different points in space. This approach is called spatial filtering. The system acts as a spatial filter that consolidates acoustic signals received by individual microphones, i.e., creates a beam; therefore, the process is called beamforming. For wide-band signals (such as speech), a wide-band beamformer is equivalent to applying finite impulse response (FIR) filters to each microphone output and then summing these signals. The coefficients of such a system are selected in such a way that the system’s performance meets a given criterion, e.g., maximizing the signal, minimizing the noise level, or eliminating it. However, the efficiency of the systems depends not only on the filter coefficients and their length but also on the array configuration. Thus, a key step in beamforming is the selection of the appropriate spatial configuration of the microphones to optimize the performance of the beamforming system [2].

The selection problem involves selecting the most appropriate set of microphones from those that make up the array. The selected set can effectively estimate the desired signal while minimizing interference from undesirable sources [3,4,5]. The selection process can significantly affect the accuracy and quality of the beamforming results. Therefore, it is important to develop robust and efficient sensor selection methods to achieve optimal beamforming performance.

In general, several approaches have been proposed in the literature for sensor selection in near-field beamforming. A common method is based on a spatial coherence array, which measures the spatial correlation between sensors. The most suitable subset of beamforming sensors with the desired properties can be identified by analyzing the characteristics of the coherence array [6,7,8].

The classical approach to the spatial filtering problem utilizes one of the standard microphone layouts (e.g., horizontal, vertical, spherical, or equally spaced in a rectangular array). It is a convenient approach since we can use a set of microphones that are mounted on a tripod (a fixed microphone array can be used). To improve the efficiency of such a system, the location of each microphone can be manipulated in one or two directions. But this solution needs some additional mechanics that allow for precise movements of particular microphones or groups of them. The other solution is based on the natural assumption that if more microphones are utilized, better results of beamforming can be reached. Thus, a microphone array would be spanned to form a larger geometrical structure. However, increasing the number of microphones is not always an optimal solution. This type of solution not only increases the cost of the array and the power consumption, but does not have to improve the desired criterion value as well. In this case of having a large microphone array, only a selected set of microphones is utilized, while others are inactive and do not participate in the beamforming. This allows one to use the same large array in a different scenario; each time, a different set of microphones gives optimal beamforming quality. This technique is called array thinning and has been successfully implemented in antenna arrays [9,10,11,12,13,14,15]. However, there are only a few papers devoted to thinning microphone arrays [16,17].

Gao et al. [18] considered a problem of sparse beamformer design in the case of the norm $L_{1}$ and, based on the properties of this norm, they were able to reduce the size of the microphone array. In [16], a thinning of a microphone array was proposed based on the Taguchi method. The authors studied the effectiveness of the Taguchi method in determining the microphone configuration.

Another approach to sensor selection in near-field beamforming involves the use of optimization algorithms such as genetic algorithms [19], particle swarm optimization [20,21,22], or simulated annealing [17,23,24]. These methods aim to find the optimal sensor combination that maximizes a beamforming performance metric, such as the signal-to-interference plus noise ratio (SINR) or beamforming gain. In recent years, machine learning techniques have also gained popularity in the selection of near-field beamforming sensors. These methods take advantage of the power of data-driven algorithms to learn patterns and relationships from training data, enabling the identification of sensors that provide the most beamforming information [25].

The problem of choosing the optimal set of microphones considered in the paper is at least NP-hard and cannot be solved optimally in polynomial time (unless P = NP). To solve this problem, metaheuristic algorithms were implemented for the antenna array [9,10] and for the microphone array [17,19], and, since this is a discrete optimization problem, an efficient branch-and-bound algorithm was developed for the signal-to-interference-plus-noise ratio (SINR) optimization criterion [26]. However, applying exact methods for different sensor selection problems is rare, and this approach has not been studied in a sufficient way. In many cases, the problem is relaxed to convex optimization problem and solved with available gradient methods or greedy algorithms [3,4].

Therefore, in this paper, a branch-and-bound algorithm, which is an exact method, is proposed to solve the microphone array thinning problem. The presented approach is universal and can be applied not only to microphone arrays, but also to antenna arrays, that is, the methodology for the generation of consecutive solutions can be applied to different types of sensor selection problem. The algorithm that generates a solution tree that must be searched using a depth search method is provided. The solution is built from scratch, and, at each step, one more microphone is turned on. The child node always has one more active microphone than its parent. The presented approach to the problem considered is new and has not yet been considered in the scientific literature.

The paper is organized as follows. Section 2 describes the problem in detail. The proposed solution is presented in Section 3, while, in Section 4, an experimental analysis of the proposed approach is examined and the results of the algorithm developed are presented. Next, in Section 5, we discuss the possibility of using the provided coding scheme in heuristic methods, called greedy algorithms. The work ends with a short summary and future research ideas in Section 6.

## 2. Problem Formulation

In this paper, we focus on the sensor selection problem for the near-field broadband beamforming problem, in which the system response should be as close as possible to the desired one. Formally, the problem considered can be defined as follows. A microphone array consisting of *N* microphones is given. The geometric center of the array is at the point denoted $r_{c}$. The microphones do not have to be equally spaced; the array can be of any shape, and there are no restrictions on microphone positions, e.g., microphones can be placed in a rectangular or spherical shape. The signals from the microphones are sampled synchronously and then the digital signals are directed to the inputs of the FIR filters of length *L* (each array element consists of the microphone followed by a *L*-tap FIR filter). At a given time, a chosen number of microphones K={1,2,…,N) can be active, while others remain inactive.

The solution can be defined by the position vector of the microphones (which follows directly from the set of active microphones λ=(λ(1),λ(2),…,λ(i),…,λ(K))). The microphone λ(i), that is, the *i*-th active microphone, has position $r_{i}$. Let Λ denote the set of all possible permutations of active and inactive microphones. There are $2^{N}-1$ possible solutions because the solution with all inactive microphones must be excluded by definition.

The transfer function of microphone λ(i) in the near field is a function of
(1)$$A\left(r_{i},f\right)=\frac{1}{||r-r_{i}\left|\right|}e^{-j2\pi f||r-r_{i}\left|\right|/c},$$
where *c* is the speed of sound in the air and *r* is the location of the sound source. The frequency response of the microphone λ(i) filter is given by
(2)$$h_{i}^{T}d_{0}\left(f\right)=H\left(h_{i}^{T},f\right),\;i=1,\ldots ,K,$$
where
(3)$$h_{i}={\left[h_{i}\left(0\right),h_{i}\left(1\right),\ldots ,h_{i}\left(L-1\right)\right]}^{T},\;h_{i}\in R^{L}$$
denotes the coefficients of the *i*-th FIR filter of length *L* and
$$d_{0}\left(f\right)=\left[1,e^{\frac{-j2\pi f}{f_{s}}},\ldots ,e^{\frac{-j2\pi f(L-1)}{f_{s}}}\right].$$

For the given set of active microphones λ, a system response can be found by solving the following equation: (4)$$G_{\lambda }\left(r,h,f\right)=\sum_{i=1}^{K}H_{\lambda }\left(h_{i},f\right)A_{\lambda }\left(r_{i},f\right)=A_{\lambda }\left(r_{i},f\right)H_{\lambda }\left(h_{i},f\right)$$
where $A_{\lambda }\left(r,f\right)=\left[A_{1}\left(r,f\right),\ldots ,A_{K}\left(r,f\right)\right]$ is a vector containing transfer functions of all microphones for set λ and $H_{\lambda }\left(h_{i},f\right)={\left[H_{1}\left(h_{1},f\right),\ldots ,H_{K}\left(h_{N},f\right)\right]}^{H}$ is the frequency filter response vector for set λ.

The desired response of the system $G_{d}\left(r,f\right)$ is also defined, where *r* is the location of the sound source and *f* is a frequency. The problem addressed in this paper is to design the microphone array (i.e., to determine the set of active microphones—their number, positions, and the FIR filter coefficients) so that the actual output of the beamformer is as close as possible to the desired one in the sense of $l_{2}$ norm.

Taking the system response $G_{\lambda }\left(r,h,f\right)$ allows one to calculate a vector of filter coefficients $h_{\lambda }$ that minimizes the objective function: (5)$$E_{\lambda }\left(r,h\right)=\frac{1}{\left||\Omega |\right|}\int_{\Omega }\sigma \left(r,f\right)||G_{\lambda }\left(r,h,f\right)-G_{d}\left(r,f\right){\left|\right|}^{2}drdf$$
where σ(r,f) is a positive weighting function, while Ω defines spatial–frequency domain. Usually, Ω consists of the region of the passband $\Omega_{P}$ and the stopband $\Omega_{S}$, that is, $\Omega =\Omega_{P}\cup \Omega_{S}$. Therefore, the problem of finding the optimal set of coefficients (frequency response) can be determined by solving the quadratic problem, which can be solved very quickly using quadratic programming techniques [27]: (6)$$\min_{\tilde{H}\in \Gamma^{N}}E\left(\tilde{H}\right)$$
where
(7)$$E\left(\tilde{H}\right)=\frac{1}{\left||\Omega |\right|}\int_{\Omega }\sigma \left(r,f\right)\left|\right|A^{H}\left(r,f\right)\tilde{H}\left(f\right)-\tilde{G_{d}}\left(r,f\right){\left|\right|}^{2}drdf,$$
and Γ=u(f)+jv(f), where u(f),v(f) are continuous and integrable, and there exist left and right derivatives such that v(0)=0 and $v(f_{s}/2)=0$.

Based on [27], one can assume that there is a performance limit for finite filter length designs, and further increasing the length of the filter does not significantly improve the criterion value. For a filter of sufficient, fixed length, one can write the beamformer design problem as
(8)$$\min_{\lambda \in \Lambda \left(M\right),h\in R^{KxL}}\left(E\left(r,h\right),\lambda \right)$$

## 3. Algorithms

Since the problem considered is at least NP-hard, there does not exist an optimal algorithm with polynomial complexity (unless P=NP). Therefore, this paper focuses on an exact algorithm belonging to the branch-and-bound (B&B) technique. It is an algorithm paradigm that must be completed for each specific type of problem, and there exist numerous choices for each of the components [28]. As B&B can only be applied to small instances, a metaheuristic hybrid algorithm is proposed to find a satisfactory solution for larger microphone arrays.

### 3.1. Branch-and-Bound

This section is devoted to the B&B algorithm for the general problem of choosing the set of active/inactive devices. The general scheme of this type of algorithm can be described as follows. The set of candidate solutions is kept as a tree with a root. The branches of this tree contain subsets of the solution set (each node is a partial solution and part of the solution set). Starting with the root, the algorithm explores the branches using a depth-first or breadth-first search of this tree.

Before the algorithm starts, a way to determine how to calculate an upper bound (UB) should be provided. An upper bound is usually a criterion value for a particular solution. At first, it is typically calculated in advance using a heuristic, and the value herein is used as the current best-known solution. If the heuristic method is not applied, UB can be set to infinity for the minimization problem. Next, for each node visited, a lower bound (LB) is calculated. The lower bound is usually a criterion value for the relaxed problem, ensuring that, for all feasible solutions, the modified function has values less than or equal to the original function, to determine whether it is worth continuing deeper into the analyzed branch or whether this branch can be excluded from further examination. If this value LB is greater than a given upper bound, then the examined subset is removed; the branch is pruned. Therefore, it is possible to reduce the number of solutions tested. The efficiency (number of nodes visited) of the B&B scheme strictly depends on the values of UB and LB.

The algorithm starts with an initial solution, calculates its criterion value $E_{0}\left(r,h\right)$, and sets $UB=E_{0}\left(r,h\right)$. The solution is then built from scratch. Initially, a microphone is added to the solution and the lower bound for this branch is calculated. If the lower bound is smaller than the upper bound, B&B progresses deeper, that is, one more microphone is added to our solution and the lower bound is calculated and compared with the upper bound value. If the lower bound is greater than the upper bound, this branch is pruned. The general scheme of B&B is shown in Figure 1 and is described in Algorithm 1.
**Algorithm 1:** Branch-and-BoundSet $UB=E(\lambda_{best})$ provided by Simulated Annealing.Start depth-first search of solution three.While the set of solutions to check is not empty:    For node *i* calculate LB according to Equation (Equation 9) and current criterion value $E_{0}\left(r,h\right)$    If $E_{0}\left(r,h\right)<UB$ set $UB=E_{0}\left(r,h\right)$ and    else if LB≥UB cut the node *i* and go to the next branch as in Figure 2    else go to node i+1Show the best solution $\lambda_{best}$

Since an amplitude criterion is considered, one can assume that the best improvement in adding one microphone is 6 dB. This estimate is based on the best-possible case. Consider the case of an array consisting of two microphones. If a narrowband signal reaches the array (for simplicity, let us assume a sinusoidal signal), then the signal in one of the channels should be delayed to equalize the phases of both signals; then, sum both recorded signals. Since the noise in both channels is not correlated, as was assumed at the beginning of the considerations, the amplitude of the sinusoidal signals will be amplified twice, while the noise amplitude will be reduced. A good estimate of the increase in SNR in this situation is 6dB. Of course, this happens in the most favorable situation because, in reality, we are dealing with wide-band signals.

The search tree is a binary tree with depth *M*, (m=1,…,M) and, at each level *m*, it has *m* active microphones. We explore our solution tree using a depth-first method, starting with the node at which all devices are inactive. We denote this level *m* of the tree as 0. Next, we continue deeper into the tree. Let us denote the solution represented by node *n* at level *m* as $\lambda_{n}$ and its criterion value as $E_{\lambda_{n}}\left(r,h\right)$. This solution can be extended by selecting more microphones; however, the number of microphones available is bounded by the number of levels below the node *n* (let us denote this number by $n_{m}$). Therefore, the LB for the given node branch is calculated as follows: (9)$$LB=E_{\lambda_{n}}\left(r,h\right)-6*n_{m}$$

For each node, its criterion value $E_{\lambda_{n}}\left(r,h\right)$ is compared with UB. If it is smaller than UB, then $UB=E_{\lambda_{n}}\left(r,h\right)$.

It is difficult to extract the exact impact of a single sensor at the criterion value and estimate LB more efficiently. An example of pruning is shown in Figure 2.


**Figure 2 sensors-24-00470-f002:**
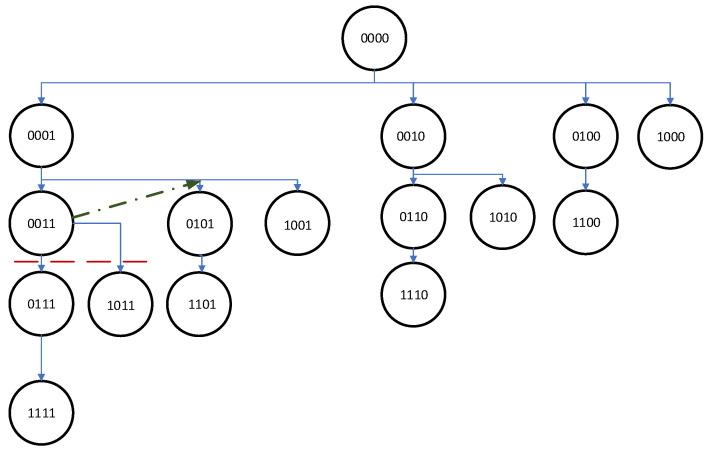
Example of solution tree for 4 microphones—pruning brunches belonging to node “0011”. The next node to check is “0101”.

#### Heuristic Algorithms

In addition to the B&B scheme, two heuristic algorithms and one hybrid metaheuristic algorithm were proposed. The first one is based on B&B scheme. The branch is pruned if adding (turning on) one additional microphone does not improve criterion value. The provided solution is a local optimum. The general scheme of the algorithm is presented below (see Algorithm 2).
**Algorithm 2:** HA1Set $crit=E(\lambda_{best})$ for criterion value calculated for the best solution with one active microphone.Start depth-first search of solution three.While the set of solutions to check is not empty:    For node *i* calculate current criterion value $E_{0}\left(r,h\right)$    If $E_{0}\left(r,h\right)<crit$ set $crit=E_{0}\left(r,h\right)$ and go to node i+1    else cut the node *i* and go to the next branch as in Figure 2Show the best solution $\lambda_{best}$

In the second proposed algorithm, the solution is built from scratch. At each step, a microphone that provides the best improvement of criterion value is chosen and added to the solution. The algorithm stops if selecting additional microphone does not improve the criterion value (greedy approach), see Algorithm 3.

Next, an algorithm proposed by Dong et al. [29] was adapted and implemented for our problem. It is a hybrid genetic algorithm that contains methods known as cuckoo search and simulated annealing. This algorithm is called NGA in the later part of this paper. As in genetic algorithms, a set of chromosomes is given, which can be called particles in some hybrid methods; these chromosomes are subjected to a crossover operation. Each chromosome represents an individual solution. The crossover operator is based on the ITÖ process and its length depends on the particle radius, the environment temperature, and the activity intensity. After the crossover is performed, new individuals (particles) are formed and the set of new solutions is checked (Algorithm 3).
**Algorithm 3:** HA2Set $crit=E(\lambda_{best})$ for criterion value calculated for the best solution with one active microphone.Add the microphone that provides the best improvement of criterion value; set $crit=E(\lambda_{best})$ for this solutionIf adding any microphone does not improve the criterion value, show the best solution $\lambda_{best}$

At the beginning, a set of *X* random particles is given. Next, these particles are classified according to the best-to-worst order and are represented by $x_{i}\in 1,\ldots ,X$, where *X* is the number of particles. In the next step, the  radius  of  the  particle for each particle $x_{i}$ is calculated as follows:(10)$$rad\left(x_{i}\right)=\frac{\left(X-i\right)\left(rad_{max}-rad_{min}\right)}{X-1}+rad_{min}$$
where $rad_{max}$ and $rad_{min}$ are maximum and minimum particle radius, respectively, and all particle radii are uniformly distributed in $rad_{max}$ and $rad_{min}$.

As in the simulated annealing algorithm, the  environment  temperature is gradually reduced and this process is defined by $T_{it}=\gamma *T_{it-1}$, where it is the number of iterations and γ<1 is a cooling coefficient.

Based on the radius of each particle, the activity  intensity of each particle $x_{i}$ is calculated. This parameter controls the intensity of the movement of the particles:(11)$$I_{i}=\frac{\left(e^{-rad_{i}}-e^{-rad_{max}}\right)}{\left(e^{-rad_{min}}-e^{-rad_{max}}\right)}e^{-1/T},$$
where $rad_{i}$ is the radius of particle $x_{i}$ and *T* is the current temperature.

To perform a crossover, a crossover  length $L_{i}$ is calculated for each particle $x_{i}$ and this value is controlled by the intensity of activity as follows:(12)$$L_{i}=\beta *I_{i}*l,$$
where β is a random number from uniform distribution and β∈[0;1]. The starting position $s_{i}$ of a crossover operator is randomly selected and the continuous positions $L_{i}$ in the particle $x_{i}$ are crossed with the best solution. Each particle codes a solution and is represented by a vector of real numbers uniformly distributed in [0;1]; the length of the vector is equal to the size of the array, that is, the number of microphones available. During decoding, the value of each element of the vector is rounded. Next, if it is equal to 1, this element is active; otherwise, it is inactive, that is, it is not chosen. Since the activity of each microphone is coded as a real number, the following crossover operator is used:(13)$$x\_new\left(i\right)=(\alpha *x_{i}\left(j\right)+\left(\left(1-\alpha \right)*best\_one\left(j\right)\right);$$
where $j=s_{i},s_{i}+1,\ldots ,s_{i}+L_{i}$, α={0;1} is a crossover coefficient, and best_one(j) is the current best solution (Algorithm 4).
**Algorithm 4:** NGA1. Define objective function f(x)=C2. Set the parameters of the algorithm.3. Initialize *n* particles.Calculate the fitness of all particles and store the best_one particleCalculate the radius of all particles using Formula (Equation 10)Calculate activity intensity using Formula (Equation 11)Sort the particles according to their fitness.Calculate the crossover length using Formula (Equation 12)While ($iter<max_{it}$):   for all particles do:       Perform the crossover operator and the mutation operator to generate a new solution   end for   Calculate the fitness for all particles and store the new_best solution   If new_best solution is better than the previous set best_one=new_best   Sort all the particles by their fitness   Update algorithm temperature   Update crossover length of all particlesReturn best_one.

## 4. Results

In this section, the results of the numerical experiments are presented. All codes are implemented in the MATLAB platform and run on a PC with Intel(R) Core i7 CPU with 2.5 GHz. Our algorithm was tested for three different regions Ω. The desired response function is defined as in [27]. This includes the frequency range of the human voice. Since we should account for the delay for the speech to reach the microphones, the desired response function in the passband region is defined as
(14)$$G_{d}\left(\lambda ,r,f\right)=e^{-j2\pi f(((||r-r_{c})/c+\left(L-1\right)/2)T)},$$
where $r_{c}=1/N\sum_{i=1}^{N}r_{i}$ denotes the center position of the placement variable λ and c=340.9 m/s is the speed of sound in air. Since the maximum frequency is chosen as 4 kHz, the sampling rate is set to 8 kHz. The weighting function is chosen as σ(r,f)=1. According to [27], the maximum filter length was set as L=40.

We examined our algorithm for three different $\Omega =\{\Omega_{1},\Omega_{2},\Omega_{3}\}$. For $\Omega_{1}$, the passband region is defined as follows: $$\Omega_{1p}=\left\{\left(r,f\right):\;-0.4\;m\leq \;|x|\;\leq 0.4\;m,y\;=\;0\;m,0.5\;\mathrm{kHz}\leq \;f\;\leq 1.5\;\mathrm{kHz}\;\right\}$$
and the stopband is
$$\begin{array}{c} \Omega_{1s}=\left\{\left(r,f\right):\;-3.0\;m\leq \;|x|\;\leq 3.0\;m,y\;=\;0\;m,2.0\;\mathrm{kHz}\;\leq f\;\leq 4.0\;\mathrm{kHz}\right\}\\ \cup \{(r,f):\;1.8\;m\leq \;|x|\;\leq 3.0\;m,y\;=\;0\;m,0.5\;\mathrm{kHz}\leq f\leq 1.5\;\mathrm{kHz}\}\\ \cup \{(r,f):-3.0\;m\leq |x|\leq -1.8\;m,y=0\;m,0.5\;\mathrm{kHz}\leq f\leq 1.5\;\mathrm{kHz}\} \end{array}$$

For $\Omega_{2}$, the passband region is defined as follows: $$\Omega_{2p}=\left\{\left(r,f\right):\;-0.4\;m\leq \;|x|\;\leq 0.4\;m,y\;=\;0\;m,0.5\;\mathrm{kHz}\leq \;f\;\leq 1.5\;\mathrm{kHz}\;\right\}$$
and the stopband is
$$\begin{array}{c} \Omega_{2s}=\left\{\left(r,f\right):\;1.8\;m\leq \;|x|\;\leq 3.0\;m,y\;=\;0\;m,0.5\;\mathrm{kHz}\leq f\leq 1.5\;\mathrm{kHz}\right\}\\ \cup \{(r,f):-3.0\;m\leq |x|\leq -1.8\;m,y=0\;m,0.5\;\mathrm{kHz}\leq f\leq 1.5\;\mathrm{kHz}\} \end{array}$$

For $\Omega_{3}$, the passband region is defined as follows: $$\Omega_{2p}=\left\{\left(r,f\right):\;-0.4\;m\leq \;|x|\;\leq 0.4\;m,y\;=\;0\;m,0.5\;\mathrm{kHz}\leq \;f\;\leq 1.5\;\mathrm{kHz}\;\right\}$$
and the stopband is
$$\begin{array}{c} \Omega_{3s}=\left\{\left(r,f\right):\;1.8\;m\leq \;|x|\;\leq 3.0\;m,y\;=\;0\;m,0.5\;\mathrm{kHz}\leq f\leq 1.5\;\mathrm{kHz}\right\} \end{array}$$

The desired responses for $\Omega_{1},\Omega_{2},\Omega_{3}$ are presented in Figure 3, Figure 4 and Figure 5, respectively.

The placement feasible region is
Λ={(x,y):−0.1m≤x≤0.1m,0.9m≤y≤1.5m}
for an array of maximum size 4 × 4 microphones and
Λ={(x,y):−0.15m≤x≤0.15m,0.9m≤y≤1.5m}
for arrays of size 5 × 5 and 6 × 6.

Obviously, the space of possible microphone locations cannot contain the location of the sound source. The numerical experiment starts with the microphone array that is, at the beginning, equispaced and fills the entire region. Both the passband and stopband are discretized, the frequency points are taken every 0.1 kHz, and the spatial points are taken every 0.1 m.

It should be noted that the running time of the algorithm for only one array configuration is long. It takes ca. 0.18 s for an array of four microphones and increases with the number of microphones.

### 4.1. Branch-and-Bound

In this section, the results of numerical experiments for the B&B algorithm are presented. As an upper bound, a value of the criterion of the solution provided by simulated annealing is taken [23]. This easy-to-implement, local search metaheuristic is fast and efficient for many discrete optimization problems. Its main steps are presented below (Algorithm 5). The number of iterations is indicated by $max_{i}t$; *T* and γ are the initial temperature and cooling parameters, respectively. In the implemented version of this algorithm, the number of iterations strictly depends on the size of the problem. Due to the long running time, it is limited to the half-size of the solution space: $0.5\times 2^{N}$, where *N* denotes the number of microphones in the array. For a 12-microphone array, the number of SA iterations was reduced to 500.

As the UB is calculated, the B&B algorithm is run. We ran the algorithm 20 times. The results of B&B are collected in Table 1. In the second column, the microphone configuration is shown (rows and columns). In the third row, the mean and minimum criterion values of UB (in brackets) are collected. These values were provided by a simulated annealing algorithm. Next, in the fourth column, the optimal criterion value is presented for the given configuration. The fifth column consists of the mean and minimum number (in brackets) of nodes that were visited, that is, solutions that were checked. The mean running time of the B&B algorithm is presented in the last column.
**Algorithm 5:** Simulated annealingDefine objective function $\min_{\lambda \in \Lambda \left(M\right),h\in R^{KxL}}\left(E\left(r,h\right),\lambda \right)$Calculate criterion for the full active array $\lambda_{f}$, set *T*, $max_{it}$ and γ, $E_{best}=E\left(\lambda_{f}\right)$Generate a random initial solution $\lambda_{a}$, $E_{curr}=E\left(\lambda_{a}\right)$While ($iter<max_{it}$) or (stop criterion)    Choose $\lambda_{anew}$ by a random switch of two microphone activeness and negate    activeness of two random microphones    Assign $\lambda_{a}=\lambda_{anew}^{\prime }$ with probability    $P\left(T,\lambda_{a},\lambda_{anew}\right)=\min\left\{1,\exp\frac{-(E\left(\lambda_{anew}\right)-E\left(\lambda_{a}\right)}{T}\right\}$    If $E\left(\lambda_{anew}\right)<E_{best}$$\lambda_{best}=\lambda_{anew}$    $T=\frac{T}{1+\gamma T}$Show the best solution $\lambda_{best}$ and $E_{best}$

The example of the optimal solution (set of active microphones) and its response are presented in Figure 6 and Figure 7, respectively.

### 4.2. Heuristic Algorithms

The results presented in Table 1 show that the proposed algorithm B&B reduces the calculation time and the number of nodes visited; however, its running time is long and increases exponentially. Therefore, two greedy algorithms and a metaheuristic approach based on the hybrid genetic method proposed in [29] were examined. The parameters of the algorithm were chosen experimentally. The number of particles is equal to the number of microphones. The initial parameters of the NGAs are as follows: the initial environment temperature is 1000, the total number of iterations (stopping condition) is the number of microphones multiplied by 3, $rad_{max}=1$, $rad_{min}=0$, cooling coefficient γ=0.9, and crossover coefficient α=0.1. The results of the proposed metaheuristic algorithm are collected in Table 2. For each instance, the algorithm was executed 20 times. The second column contains the microphone configuration (rows and columns). In the third row, the mean and minimum criterion values (in brackets) are gathered. The mean running time of the algorithm NGA is presented in the last column. Results of algorithms HA1 and HA2 are gathered in columns 5–6 and 7–8, respectively.

## 5. Discussion

The problem considered is important from a practical point of view. However, there is a lack of papers devoted to exact algorithms and only some heuristic methods have been proposed in the literature. However, without knowing the optimal criterion value, it is hard to verify effectiveness. During the construction of the B&B algorithm, the main problem is to extract the impact of a single microphone on the value of the criterion. Therefore, it was difficult to construct a more efficient lower bound that works for the general case of the problem considered. The proposed pruning condition cuts roughly 10–20% of the solutions. Finding an optimal set of filter coefficients (frequency response) and calculating the criterion value depend on the number of microphones and take a long time (0.15 s for microphone placement), even for small arrays (4 elements). Therefore, even such a reduction is significant. The number of visited nodes does not depend on the defined passband and stopband regions; therefore, we can conclude that the proposed method is robust. It can be seen that the number of visited nodes depends on the initial value of the upper bound UB; however, the improvement is more visible for larger instances of the problem.

The proposed metaheuristic approach can find an optimal solution for smaller instances (for an array of four and six microphones, it found the optimal value of the criterion almost every time). For a greater number of microphones, the metaheuristic approach is able to provide satisfying criterion values. However, to achieve a good solution, the running time of the algorithm should be long and the number of iterations should depend on the size of the array.

The HA1 method is efficient only for small instances. Its running time is short, and results are close to optimal. However, for bigger instances (5 × 5 and 6 × 6), the running time of the algorithm was unacceptable, and it stopped after 10 min. The criterion values provided by HA2 are comparable with those of HA1 and NGA. Its running time is the shortest. In the case of instances with a greater number of sensors, the greedy approach is the most efficient. It can provide a good solution in a short time. For smaller instances, NGA was the best one and its solutions were close or even globally optimal.

Future work should focus on answering the following questions. Is it possible to provide a better estimate of LB? Is it possible to extract the impact of a specific, single microphone on a value of the criterion? Is it possible to provide elimination procedures that significantly reduce the number of visited nodes?

## 6. Conclusions

In this paper, a general branch-and-bound algorithm was proposed for the sensor selection problem. During the depth-search method, successive solutions in one branch differed only in one bit, which is equivalent to microphone activation/disactivation. The proposed general pruning scheme decreases the number of nodes visited (checked solutions). Then, an efficient metaheuristic approach was proposed and examined during numerical experiments. Future work will focus on providing efficient elimination procedures for the branch-and-bound algorithm.

## Figures and Tables

**Figure 1 sensors-24-00470-f001:**
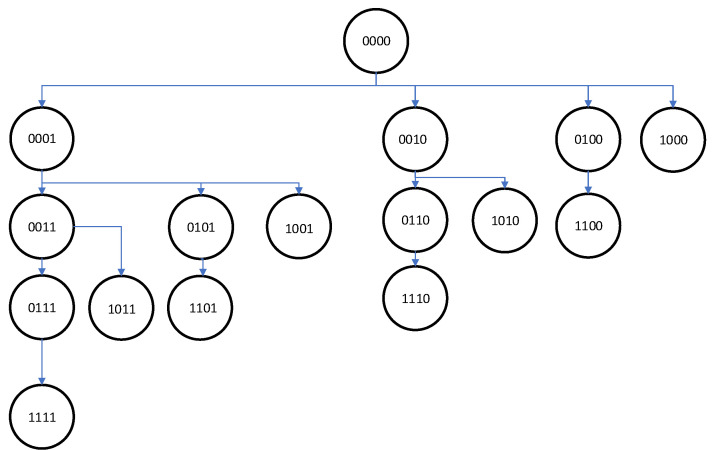
Example of solution tree for 4 microphones; 1 denotes an active microphone, while 0 is an inactive microphone.

**Figure 3 sensors-24-00470-f003:**
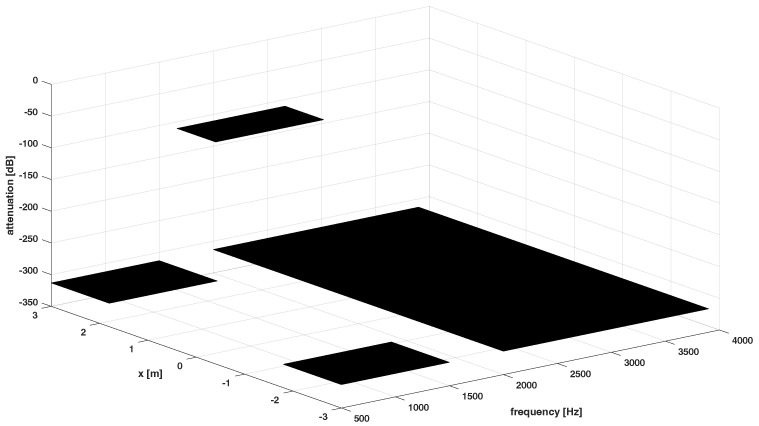
Desired response of the system−the dark spaces denote stopband regions, the lighter space denotes passband region for $\Omega_{1}$.

**Figure 4 sensors-24-00470-f004:**
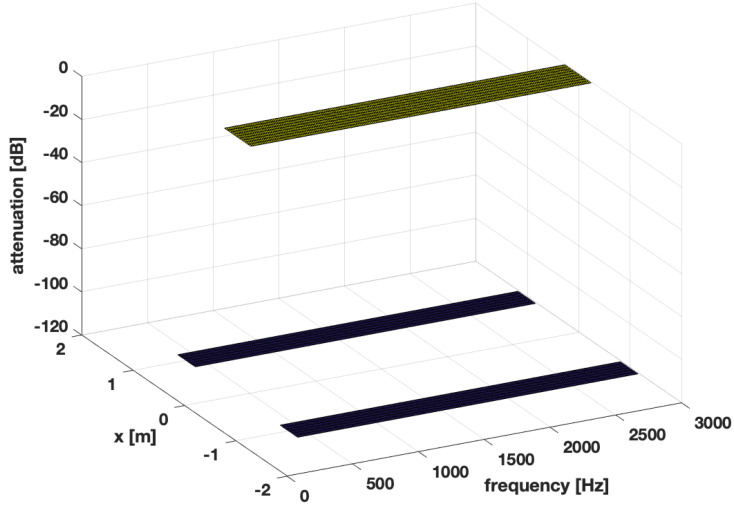
Desired response of the system−the dark spaces denote stopband regions, the lighter space denotes passband region for $\Omega_{2}$.

**Figure 5 sensors-24-00470-f005:**
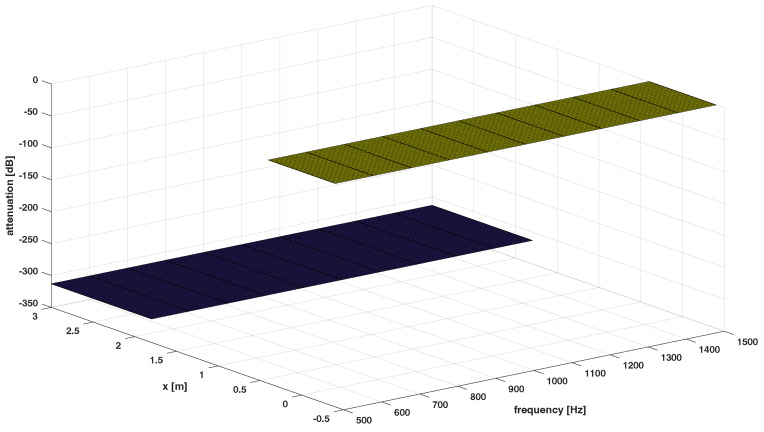
Desired response of the system−the dark spaces denote stopband regions, the lighter space denotes passband region for $\Omega_{3}$.

**Figure 6 sensors-24-00470-f006:**
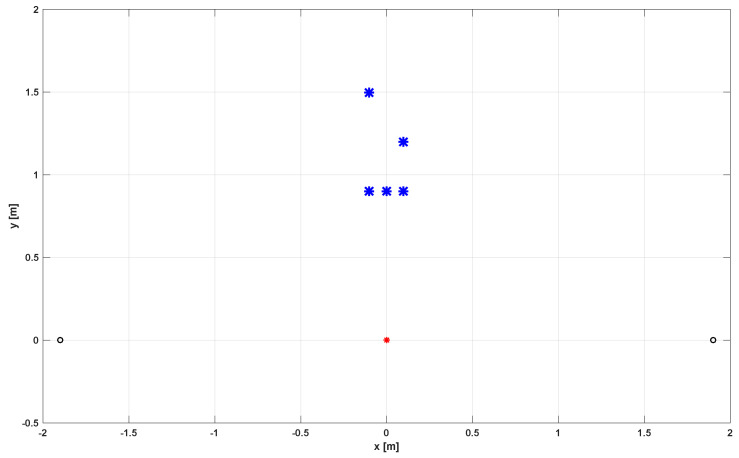
The set of selected sensors for array 3 × 3 and $\Omega_{1}$.

**Figure 7 sensors-24-00470-f007:**
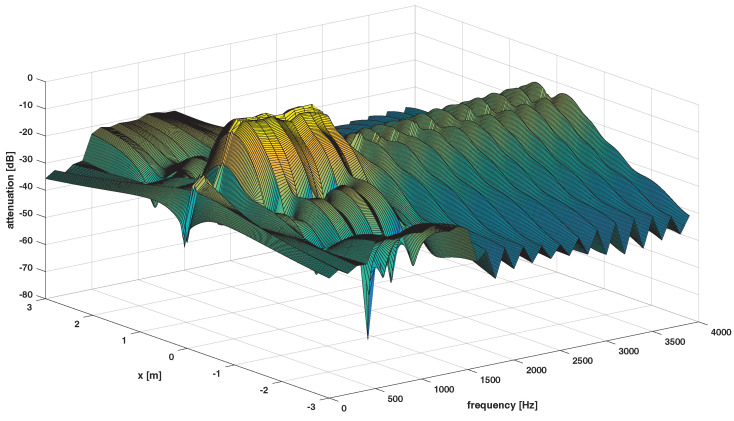
Optimal system response for array 3 × 3 and $\Omega_{1}$.

**Table 1 sensors-24-00470-t001:** The results of B&B algorithm.

Space	Microphone Configuration	${\mathrm{UB}}_{\mathrm{in}}$ [dB]	B&B Crit. [dB]	No. of Solutions	Running Time [s]
$\Omega_{1}$	2 × 2	−38.89 (−38.89)	−38.89	13 (13)	0.94
	3 × 2	−36.34 (−38.60)	−38.60	56 (51)	3.71
	4 × 2	−37.69 (−38.85)	−38.85	233 (222)	19.39
	3 × 3	−40.64 (−40.68)	−40.68	472 (472)	36.80
	5 × 2	−40.40 (−40.60)	−40.72	894 (882)	69.22
	6 × 2	−39.52 (−40.66)	−40.83	3842 (3718)	320.33
$\Omega_{2}$	2 × 2	−25.64 (−25.64)	−25.64	13 (13)	0.92
	3 × 2	−26.03 (−28.34)	−28.34	54 (50)	3.76
	4 × 2	−27.49 (−28.60)	−28.60	232 (222)	16.88
	3 × 3	−30.16 (−30.42)	−30.42	475 (472)	36.60
	5 × 2	−29.87 (−30.35)	−30.47	912 (882 )	66.44
	6 × 2	−30.27 (−30.51)	−30.57	3747 (3706)	318.14
$\Omega_{3}$	2 × 2	−21.83 (−21.83)	−21.83	13	0.95
	3 × 2	−22.23 (−24.54)	−24.54	56 (51)	4.05
	4 × 2	−24.12 (−24.79)	−24.79	226 (222)	Data
	3 × 3	−26.45 (−26.62)	−26.62	475 (422)	36.13
	5 × 2	−25.88 (−26.66)	−26.66	936 (876)	72.81
	6 × 2	−26.20 (−26.70)	−26.76	3767 (3706)	295.42

**Table 2 sensors-24-00470-t002:** The results of the NGA algorithm.

Space	Microphone Configuration	Criterion Value [dB] NGA	Running Time [s] NGA	Criterion Value [dB] HA1	Running Time [s] HA1	Criterion Value [dB] HA2	Running Time [s] HA2
$\Omega_{1}$	5 × 2	−40.57 (−40.72)	29.65	−40.60	25.03	−39.62	3.55
	6 × 2	−40.77 (−40.83)	40.13	−40.49	40.58	−40.49	5.29
	7 × 2	−40.26 (−40.42)	57.26	−40.27	36.30	−40.22	7.00
	4 × 4	−41.53 (−41.89)	77.65	−41.86	53.45	−40.82	5.01
	5 × 5	−41.75 (−41.78)	222.27	-	-	−41.61	6.05
	6 × 6	−40.58 (−41.34)	541.30	-	-	−41.77	11.88
$\Omega_{2}$	5 × 2	−30.16 (−30.47)	29.57	−30.34	24.08	−29.41	3.44
	6 × 2	−30.17 (−30.57)	38.90	−30.24	39.01	−30.24	4.86
	7 × 2	−29.91 (−29.97)	55.77	−30.01	33.59	−29.96	6.06
	4 × 4	−31.50 (−31.70)	72.31	−31.60	42.46	−30.57	4.85
	5 × 5	−31.42 (−32.08)	214.17	-	-	−31.35	6.25
	6 × 6	−30.35 (−31.04)	538.07	-	-	−31.52	11.41
$\Omega_{3}$	5 × 2	−26.55 (−26.66)	28.70	−26.53	26.49	−25.61	3.45
	6 × 2	−26.45 (−26.77)	43.32	−26.43	40.65	−26.43	4.41
	7 × 2	−26.41 (−26.46)	57.72	−26.20	34.64	−26.15	6.04
	4 × 4	−27.79 (−28.04)	73.65	−27.80	43.76	−26.76	4.64
	5 × 5	−28.17 (−28.56)	187.45	-	-	−27.54	6.04
	6 × 6	−27.71 (−28.01)	511.12	-	-	−27.71	13.24

## Data Availability

Data are contained within the article.

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
