# Peer review of "A General Scheme of a Branch-and-Bound Approach for the Sensor Selection Problem in Near-Field Broadband Beamforming"

_sensors, 2024, doi:10.3390/s24020470_

Round 1

Reviewer 1 Report

Comments and Suggestions for Authors

This work proposes a general scheme for sensor selection in near-field broadband beamforming. Overall, the paper the clearly written and the topic is interesting. Some minor comments are provided below:

-- Authors are encouraged to discuss the feasibility of using GA in pratical applications. 

-- Comparison with benchmark methods can be better highlighted. 

Comments on the Quality of English Language

reads conformatable 

Author Response

Dear reviewer, thank you for your valuable comments. Please see the attachment.

Reviewer 2 Report

Comments and Suggestions for Authors

This paper presents a branch-and-bound method for the microphone array topology design problem, which is important in the context of broadband beamforming. I found some problems in the current form that must be addressed at this stage.

1. The novelty and contribution of this work should be more clearly stated, as the problem formulation is completely the same as what Nordholm did in [21] and the B&B algorithm was widely used in many fields. So what is the new insight of this paper.

2. The sensor selection or microphone subset selection problem was broadly investigated in literature, the author's literature review is far from sufficient. Please cite the following papers [A-E]. In the fourth paragraph of introduction, please add citations to 'the typical approach', 'in one method' and 'in another method'...

3. The experimental comparison with state-of-the-arts is not enough. Please compare the proposed method with the quadratic programming method in [21], convex optimization method in [C] and/or the greedy method in [B].

4. There are some minor typos in the paper, e.g., in abstract "in sens of", a transpose is required for the definition of d_0(f) on page 3, "The search tree is a binary tree of height M", where "height" should be depth; below equation (10) there must be something wrong with the definition of r_c. I recommend using bold symbols to denote vectors or matrices, and normal symbols for scalars.

[A] Joshi S, Boyd S. Sensor selection via convex optimization[J]. IEEE Transactions on Signal Processing, 2008, 57(2): 451-462.

[B] Shamaiah M, Banerjee S, Vikalo H. Greedy sensor selection: Leveraging submodularity[C]//49th IEEE conference on decision and control (CDC). IEEE, 2010: 2572-2577.

[C] Zhang J, Chepuri S P, Hendriks R C, et al. Microphone subset selection for MVDR beamformer based noise reduction[J]. IEEE/ACM Transactions on Audio, Speech, and Language Processing, 2017, 26(3): 550-563.

[D] Zhang J, Tao R, Du J, et al. Energy-efficient sparsity-driven speech enhancement in wireless acoustic sensor networks[J]. IEEE/ACM Transactions on Audio, Speech, and Language Processing, 2022, 31: 215-228.

[E] Zhang J, Zhang G, Dai L. Frequency-invariant sensor selection for MVDR beamforming in wireless acoustic sensor networks[J]. IEEE Transactions on Wireless Communications, 2022, 21(12): 10648-10661.

[F] Zhang J, Du J, Dai L R. Sensor selection for relative acoustic transfer function steered linearly-constrained beamformers[J]. IEEE/ACM Transactions on Audio, Speech, and Language Processing, 2021, 29: 1220-1232.

Comments on the Quality of English Language

see above

Author Response

We are very grateful for valuable comments. Please see the attachment.

Round 2

Reviewer 2 Report

Comments and Suggestions for Authors

The revised version is fine from my side. Thanks to the authors by taking my comments into account.